# Home Healthcare Among Aging Migrants: A Joanna Briggs Institute Scoping Review

**DOI:** 10.3390/healthcare13080863

**Published:** 2025-04-10

**Authors:** Areej Al-Hamad, Yasin M. Yasin, Lujain Yasin, Grace Jung

**Affiliations:** 1Daphne Cockwell School of Nursing, Toronto Metropolitan University, Toronto, ON M5B 2K3, Canada; lujain.yasin@torontomu.ca (L.Y.); yeeun.jung@torontomu.ca (G.J.); 2Faculty of Nursing, University of New Brunswick, Fredericton, NB E3B 3X9, Canada; yasin.yasin@unb.ca

**Keywords:** aging migrants, JBI, scoping review, home healthcare

## Abstract

**Background/Objectives:** The aging migrant population faces unique healthcare challenges due to linguistic, cultural, and systemic barriers. Home healthcare services play a crucial role in supporting aging migrants, yet accessibility and effectiveness remain inconsistent across different healthcare systems. This scoping review examines the experiences of aging migrants in home healthcare settings and explores the impact of these services on their health and well-being. This review aims to synthesize the existing literature on home healthcare experiences among aging migrants, highlighting the facilitators and barriers to effective service delivery and the implications for policy and practice. **Method:** A scoping review was conducted using the Joanna Briggs Institute (JBI) framework. A comprehensive search was performed across multiple databases, including CINAHL, Medline, and Scopus, for articles published between 2000 and 2024. Studies were selected based on predefined inclusion criteria focusing on home healthcare experiences among aging migrants. Data extraction and thematic analysis were conducted to identify key themes. **Results:** The review identified 35 studies across various geographical regions, highlighting four key themes: (1) Cultural and Linguistic Accessibility, (2) The Role of Informal Caregiving, (3) Structural and Systemic Challenges, and (4) Health Outcomes and Identity Preservation. The findings indicate that language barriers, cultural stigma, and systemic exclusion significantly hinder equitable access to home healthcare. Informal caregiving by family members remains a primary support mechanism, though it places considerable strain on caregivers. The lack of culturally competent healthcare services and inadequate policy frameworks exacerbate disparities in care. **Conclusions:** This review highlights the critical need for systemic reforms to improve healthcare accessibility for aging migrants. Policies must prioritize cultural competence training for healthcare providers, expand multilingual healthcare services, and integrate informal caregiving into formal support structures. Investment in community-driven healthcare initiatives and targeted outreach programs can help bridge existing service gaps. While home healthcare plays a vital role in supporting aging migrants, structural inequities and cultural barriers continue to hinder equitable access. Addressing these disparities requires comprehensive policy interventions, enhanced provider training, and culturally inclusive care models. Future research should explore innovative frameworks that incorporate culturally responsive practices to ensure effective and equitable home healthcare for aging migrant populations.

## 1. Introduction

Over the last ten years, the number of migrants around the world has more than tripled [1]. The global diaspora of migrants can be attributed to numerous factors, including religious persecution, human rights violations, and economic instability, among others [2]. As a result, many migrants leave their home countries in search of safety and opportunities to achieve economic capital and stability [2]. The global population of individuals aged 60 and over has doubled since 1980 and is expected to reach 2 billion by 2050 [3]. In light of this demographic shift, the WHO launched its “Aging Well” initiative [4], which highlights the importance of accessible and equitable care, particularly home healthcare, as a key component in supporting older adults to age with dignity and independence and other countries like Canada emphasize housing strategies to facilitate aging in place [5]. The urgency of these strategies became evident during the COVID-19 pandemic, during which many older migrants were isolated or losing their health in care facilities [6].

The process of aging and integrating into new communities presents distinct difficulties for older migrants [7]. Through resettlement in new countries, aging migrants are faced with various difficulties when adapting to their new environments, including language barriers [8], cultural differences [9], lack of employment opportunities [10], unfamiliarity with the healthcare system, and discrimination [11,12]. Furthermore, aging migrants have a lack of social networks, and when combined with a language barrier, this can lead to social isolation, loneliness, and mental health challenges [13]. Aging migrants from diverse cultural backgrounds often face cross-cultural dissonance upon arrival to their settlement countries due to differences in cultural and social norms [14]. Such cross-cultural dissonance extends to healthcare services, where differences in health beliefs, medical practices, or cultural stigmatizations can hinder aging migrants from utilizing health services [9,11].

The reluctance to use healthcare services may also extend to the use of institutional care, such as nursing homes and retirement home facilities [15]. The availability and use of social care vary according to the different cultural, social, and economic norms within in each country [9]. For example, in Arab countries, older adult care is predominantly provided by family members, as it is considered a family responsibility rather than a societal one [16]. Similar sentiments were observed in Asian and Latino cultures as they emphasize the importance of family unity and the duty to care for elders within multigenerational households [17]. Consequentially, the reluctance to engage with institutionalized healthcare services may also be linked to the norms and practices which aging migrants from diverse cultures are familiar with from their home countries [11]. Owing to their challenging and frequently traumatic pasts, migrants are perhaps mistrustful of unfamiliar institutional settings or providers [18].

The fear of losing cultural connections, which results in cultural bereavement, and the comfort of familiar surroundings may further strengthen the preference to age at home [16]. Opportunities and access to safe spaces for practicing their faith can be especially crucial for aging migrants [19]. Likewise, the availability of culturally diverse food options may be sparse in institutional settings and therefore contribute to unmet needs and preference to age at home [16]. Furthermore, migrants who follow Islamic dietary laws must consume only Halal foods, which are not always available but are critically important for the cultural identity of some older migrants and refugees [17]. These considerations highlight the need for culturally competent care, which is often lacking and can prevent aging migrants from diverse cultures from accepting institutionalized healthcare services [18]. Aging with vitality is possible, but it often involves health challenges and a greater need for daily support [20]. While age-related conditions may be unavoidable, age-friendly policies can help older adults including aging migrants maintain well-being and age at home [21]. Unlike locals, aging migrants must adapt quickly to unfamiliar environments with limited resources and support systems [19,21]. In many home countries, elderly care is provided by extended family, but in resettlement contexts, families often face their own integration challenges, making this difficult [20]. Our ongoing research with older refugees from Ukraine, Syria, and Afghanistan reveals a strong reluctance to use formal elderly care services [22].

As noted above, many cultures strongly favor family-based care, reflecting a normative expectation that children should care for their aging parents [14,17,21]. This cultural discrepancy underscores a pressing need for innovative, culturally congruent aging-in-place solutions that respect and integrate these values. Home healthcare as an informal mode of caregiving is often expected to provide the majority of emotional and caregiving assistance to aging migrants within many cultural contexts, including Middle Eastern and African communities, among others [16,23]. This informal caregiving, while crucial, often lacks the necessary formal support systems, training, and resources to meet the complex needs of aging migrants. Additionally, there remains a significant gap in knowledge regarding the specific experiences, practices, and challenges of home healthcare within these communities, as well as its impact on the health and well-being of aging migrants. Addressing this gap is essential for developing effective, inclusive, and culturally appropriate models of care that can enhance the quality of life for this vulnerable population [16,23].

## 2. Objective/Research Question

This scoping review aims to answer the following question “What is currently known in the literature about home healthcare experiences among aging migrants and the influence on their health and well-being?”

## 3. Methods

This review was conducted using the Joanna Briggs Institute (JBI) framework for scoping reviews [24] and adhered to the guidelines outlined in the Preferred Reporting Items for Systematic Reviews and Meta-Analyses extension for Scoping Reviews (PRISMA-ScR) [25].

### 3.1. Inclusion and Exclusion Criteria

The JBI PCC (Population, Concept, and Context) approach for scoping reviews was utilized to establish the inclusion and exclusion criteria, ensuring a structured focus on the population of interest, the central concept being explored, and the relevant contextual settings [24] (see Table 1 for inclusion and exclusion criteria).

### 3.2. Search Strategy

The search strategy was designed in collaboration with a research librarian with the incorporation of a comprehensive range of relevant keywords and index terms, initially tailored to CINAHL, as detailed in Appendix A The strategy was then adapted for other databases, accounting for variations in Boolean operators, truncation, and wildcard usage. A peer review of the strategy was conducted by a second librarian using the PRESS Peer Review Strategy [26]. The review targeted studies published in English from 2000 to 2024, with reference lists of selected sources screened for additional relevant studies. The 2000 cut-off was chosen due to notable policy changes and the global rise in displacement, which led to a surge in research on migrants’ healthcare needs. This timeframe ensured the inclusion of up-to-date, comprehensive, and relevant findings. The databases searched included CINAHL, Humanities Source Ultimate, Medline, Social Sciences Full Text, Scopus, Web of Science and ProQuest Dissertations and Theses. Additional searches were performed using Google Scholar and by examining reference lists from selected articles. Gray literature, such as conference proceedings and opinion papers, was also included to ensure a thorough and comprehensive review of all pertinent information [27].

### 3.3. Study Selection and Extraction

After completing the search, all records/citations were compiled and imported into EndNote v.21, where duplicates were removed. Two independent reviewers (GY and LY) initially screened the titles and abstracts of the remaining sources to determine their relevance to the review’s inclusion criteria. Relevant studies were retrieved, and their details were transferred to JBI SUMARI for centralized management, assessment, and analysis [28]. Titles and abstracts were again screened by the two reviewers (GY and LY) against the inclusion criteria, followed by a full-text review of selected studies. For studies excluded at the full-text stage, reasons for exclusion were documented and reported. Any disagreements during the selection process were resolved through discussion or consultation with an additional reviewer (YY). The entire selection process was documented using a PRISMA flow diagram (See Figure 1). Data extraction for the included studies was conducted using a customized table designed to align with the review question (Appendix A). This process utilized a standardized JBI SUMARI data extraction tool, with minor modifications to suit the review’s objectives [28]. Extracted data included details about participants, concepts, context, methodologies, and key findings. Discrepancies between reviewers were resolved through discussion or with input from another reviewer (YY). When data were missing or unclear, the team contacted study authors for clarification [24].

### 3.4. Data Analysis

The data analysis employed a basic inductive content analysis approach [29], with two independent reviewers (GY and LY) systematically documenting the characteristics of sources and the frequency of articles addressing migrants’ home healthcare, including associated challenges, practices, and outcomes. The reviewers carefully examined the collected materials, including academic studies, reports, and other relevant resources identified through the scoping review, to extract key codes related to the population, concept, and context of the study. These codes were then grouped into themes based on observed trends, patterns, and notable similarities or differences. The coding process was iterative and adaptive, allowing for continuous refinement as new data were analyzed [29]. To ensure the accuracy and consistency of the analysis, team members employed a comparative approach to resolve discrepancies, either through discussion and consensus or by consulting a third reviewer (YY) when needed (see Appendix A for the extraction table).

## 4. Results

The initial search yielded 381 records. After removing 145 duplicates and excluding 3 incomplete information records, 233 records remained for preliminary screening. During the first review phase, titles and abstracts were evaluated for relevance to the study’s objectives, resulting in 47 records selected for full-text assessment. Of these, 20 were excluded for being outside the context, 9 for not aligning with the study’s concept, 1 for not fitting the population criteria, 1 for not having an eligible context, 1 for not being in English, and 1 for which the full text not found. A targeted search using Google Scholar and a manual review of reference lists from the selected articles added eight more records. Ultimately, 35 records were included in the final scoping review. The PRISMA diagram illustrating this process is available in Figure 1.

### 4.1. Included Studies

The included studies span various geographical locations and settings, offering insights into the experiences of migrants and their interactions with home healthcare services. Geographically, studies were conducted across multiple countries, including Denmark [30,31], Canada [32,33,34], Israel [35], the United States [36,37,38,39,40,41,42,43,44,45,46], Australia [47,48,49], Sweden [50,51,52,53], Belgium [54], Switzerland [55], and the Netherlands [56,57,58,59,60,61,62]. One systematic literature review [63] and one scoping review [64] were included. Urban settings were predominant, with studies often focusing on cities with significant migrant populations, such as Copenhagen [31], Stockholm [50,52], Malmö [50], Amsterdam [57], and Los Angeles [44]. The following graph displays the geographical distribution of studies (see Figure 2).

### 4.2. Cultural and Linguistic Accessibility in Home Healthcare

Studies indicate that language barriers and a lack of cultural sensitivity significantly hinder equitable access to home healthcare for aging migrants [51,59,61,62]. Several studies have shown that many older migrants experience language attrition in later life, reverting to their native language, which complicates communication with healthcare providers and contributes to misunderstandings regarding treatment plans and medication adherence [49,54,60]. Limited health literacy among aging migrants further exacerbates these challenges, making it difficult for them to navigate complex healthcare systems, understand available services, and communicate their needs effectively [31,46]. Studies have also highlighted that older migrants often rely on informal networks, such as family members, to serve as translators or cultural mediators during healthcare interactions [49,56]. However, research has found that this reliance places a significant burden on family caregivers and does not ensure accurate medical communication, raising concerns about patient autonomy and privacy [31,46].

Various studies have reported that healthcare providers who lack cultural competence may unintentionally create alienating or dismissive interactions, leading to decreased trust and engagement among migrant patients [49,54,56]. Studies have found that cultural misalignment in healthcare settings, such as a lack of recognition for religious practices, dietary restrictions, or gender preferences in caregiving, discourages many migrant families from utilizing formal healthcare services [40,64]. Findings also suggest that migrant populations may avoid seeking healthcare due to previous experiences of discrimination or cultural insensitivity in the healthcare system, which has been reported to contribute to a sense of distrust and reluctance to engage with formal healthcare providers [54,56]. Multiple studies have examined initiatives designed to improve healthcare accessibility for aging migrants through culturally and linguistically responsive programs [46,49,54]. The findings have shown that healthcare environments staffed with bilingual professionals, along with the availability of translated materials and culturally adapted communication tools, such as visual aids and cue cards, have led to increased healthcare utilization and improved patient–provider interactions [46,49,54]. Studies have also identified culturally familiar elements, including native-language resources, traditional foods, and recognition of religious practices, as factors contributing to improved trust and engagement with healthcare services [46,62]. The findings suggest that healthcare providers who share linguistic or cultural backgrounds with their patients are often able to foster stronger relationships and enhance communication, which has been linked to better healthcare outcomes for aging migrants [31,49].

Cultural stigma surrounding formal home healthcare services presents another challenge, where caregiving is viewed as a family duty, and seeking external support is perceived as neglecting filial responsibilities [44,54]. This cultural expectation prevents many families from utilizing home healthcare services, even when caregiving needs surpass their capacity [51]. Despite the documented benefits of culturally adapted care models, research highlights inconsistencies in their implementation across different healthcare systems [31,49,54,56]. Studies have reported that financial and structural limitations often prevent healthcare institutions from hiring bilingual staff, developing translated materials, or offering cultural competency training on a large scale [31,49,54,56]. Findings also indicate that healthcare systems without mandated policies for cultural competency training face delays in the adoption of culturally inclusive healthcare models [46,54]. Many studies have identified institutional resistance to change, as well as a lack of awareness regarding the specific needs of aging migrant populations, as key factors contributing to the continued disparities in healthcare access [31,46,54]. Records on healthcare service delivery models suggest that expanding partnerships between healthcare institutions, community organizations, and religious groups has improved access to culturally tailored healthcare programs [37,43,53,64]. Research has also documented the role of community health workers and cultural mediators in bridging the gap between migrant families and formal healthcare services, providing education, advocacy, and support in navigating home healthcare resources [31,56,64]. Studies have found that digital health technologies, such as telehealth platforms with multilingual capabilities, have contributed to improved healthcare accessibility for aging migrant populations [31,46].

### 4.3. The Role of Informal Caregiving

Studies indicate that informal caregiving plays a critical role in ensuring healthcare access and well-being among aging migrant populations [43,51,52]. Migrants often rely on family members and community networks for caregiving, largely due to cultural expectations emphasizing familial responsibility and mutual support [47,51]. A study’s findings highlight that strong filial norms, reinforced by religious and cultural beliefs, shape caregiving practices and discourage institutionalized care [52]. Many older migrants refuse special housing or formal care services, preferring to remain at home under the care of family members [50,51]. Additionally, migrants face significant barriers such as limited knowledge of available public care services [46], distrust and confusion about eligibility of the services in healthcare institutions [46], and fear of discrimination [49], all of which contribute to a heightened reliance on family caregiving [52].

Studies show that family caregivers provide essential support beyond basic care, including language brokering, emotional support, and healthcare advocacy [34,57]. Second-generation immigrants, in particular, often serve as intermediaries between older migrants and formal healthcare providers, ensuring access to culturally appropriate care [43,60]. However, this dependence on informal caregiving networks places substantial burdens on caregivers, who frequently struggle to balance work, daily responsibilities, and caregiving duties without adequate training or financial support [52]. The lack of formal recognition and institutional support further exacerbates caregiver stress, contributing to emotional exhaustion and financial strain [49]. Additionally, caregiving roles within migrant families are often shaped by cultural traditions, with responsibilities frequently assigned based on gender and familial hierarchy [49,57]. In many cases, women bear the primary caregiving burden, a role reinforced by traditional expectations of filial piety and reciprocal obligations [45]. However, studies suggest that migration necessitates adaptations to these caregiving structures, leading to a blended model of care that integrates family support with limited external assistance [50,51,52]. While caregiving is often perceived as an act of love and gratitude, findings indicate that caregivers experience significant stress, particularly when formal support systems are inadequate [50,51,52].

Studies also demonstrate that these networks help mitigate language and cultural barriers by facilitating communication with healthcare providers and advocating for culturally appropriate services [47,51]. However, caregiving responsibilities extend beyond traditional definitions, requiring substantial emotional labor and complex decision-making [57]. Findings suggest that despite their importance, informal caregivers often operate without access to respite care, formal training, or financial compensation, leading to increased caregiver burden and negative health outcomes [43,62]. Cultural norms and systemic barriers shape informal caregiving experiences as migrant families are less likely to seek formal support due to fears of institutional neglect or culturally inappropriate care [54,56]. Studies further suggest that healthcare professionals may lack cultural competence, leading to interactions that alienate migrant families and discourage engagement with healthcare systems [31,49]. Findings also emphasize the need for culturally responsive healthcare policies that recognize the contributions of informal caregivers while ensuring equitable access to formal support services [47,51]. Community-based programs and migrant organizations enhance caregiver well-being and patient outcomes through collaboration with healthcare institutions and culturally competent services [43]. Findings also suggest that strengthening partnerships between migrant organizations and healthcare agencies can facilitate trust-building and enhance access to necessary services [57]. Studies highlight the effectiveness of initiatives that provide targeted caregiver training, financial assistance, and culturally tailored healthcare resources [43,46,52,60]. Integrating informal caregiving networks into formal healthcare systems improves accessibility for aging migrants. Policies on financial support, training, and culturally sensitive care reduce caregiver burden and sustain these networks [47,50,51]. Targeted interventions addressing language barriers [54], cultural stigma [57], and discrimination [54] improve formal and informal care models, and recognizing informal caregivers’ contributions is key to sustaining effective care for aging migrants [31,56].

### 4.4. Structural and Systemic Challenges

Structural and systematic barriers significantly hinder the utilization of home healthcare services among migrant populations [39]. Limited awareness of available services [31,54], affordability issues [19,35], and low health literacy [35] contribute to underutilization, particularly among older migrants [35,39]. Inadequate outreach efforts and the lack of multilingual informational materials further exacerbate the problem, leaving many migrants unaware of their eligibility for home healthcare [51,54]. Studies indicate that linguistic barriers are a primary challenge, as healthcare systems often fail to provide culturally and linguistically tailored information, preventing migrants from fully understanding the services available to them [19,44,49]. Affordability remains a critical concern for migrants [63], while some home healthcare services are publicly funded [52], additional costs, including transportation, caregiving supplies, and unpaid leave for family caregivers, create significant financial strain [51,54]. A study highlights systemic biases in needs assessments, where assumptions around family caregiving responsibilities often result in reduced support for migrant families [49]. In the Netherlands, policies prioritizing cost efficiency over culturally appropriate care have further widened disparities for older migrants [56]. Chaouni, Smetcoren [54] concluded that financial inequities require targeted funding initiatives and economic support programs designed to alleviate the burden on migrant caregivers.

Linguistic and logistical barriers also impede healthcare access as many migrants struggle to navigate complex healthcare systems due to limited fluency in the dominant language [49,51]. Additionally, healthcare providers often lack cultural competence, limiting their ability to effectively communicate with non-native speakers [54]. Logistical challenges, including transportation difficulties [36] and bureaucratic complexities [19], further discourage healthcare engagement, particularly among low-income and physically limited older adults [44,51]. A study suggests that culturally tailored transportation services, such as those implemented in diverse adult day centers, can improve healthcare accessibility [42].

Mistrust of institutional systems, rooted in historical inequities and perceived discrimination, further discourages engagement with formal care programs [31,49]. Studies indicate that ethnically profiled healthcare services, which integrate cultural and linguistic adaptations, significantly enhance service trust and engagement [50].

Systemic inequities within healthcare policies reinforce these barriers, disadvantaging migrants as rigid assessment frameworks often fail to accommodate cultural variations in care expectations, leading to inadequate resource allocation [53,54]. Research highlights that care managers and healthcare professionals may unintentionally reinforce cultural stereotypes, resulting in care plans that do not align with the needs of migrant families [31]. Furthermore, healthcare policies frequently overlook the specific needs of aging migrants, limiting their ability to access appropriate services [54]. Evidence from Sweden suggests that decentralized healthcare models incorporating private providers offering culturally adapted services improve accessibility and equity for older migrants [50]. These findings emphasize the need for targeted public health campaigns to increase awareness of home healthcare services among migrant communities [49,51]. Providing informational materials in multiple languages and culturally appropriate formats has been shown to improve service utilization [45,52,54]. Economic support programs tailored to migrant families can reduce financial barriers, ensuring equitable access to care [54].

### 4.5. Health Outcomes and Identity Preservation

Studies indicate that community-directed programs supporting home healthcare are associated with increased patient satisfaction [52], reduced caregiver burden [54], and better health outcomes [43]. These programs provide immediate relief to family caregivers, allowing them to manage their responsibilities more effectively while ensuring older adults receive adequate care [64]. Additionally, home healthcare fosters independence and autonomy in older adults, contributing to successful aging, positive self-identity, and healthier lifestyles [34,45]. However, a study shows that older adults receiving home healthcare for conditions such as depression often do not undergo appropriate mental health screenings, as home healthcare nurses may lack the necessary training or confidence to assess psychological conditions [43]. Additionally, existing screening tools often fail to account for cultural variations in symptom expression, reducing their effectiveness in identifying mental health concerns [43].

Studies report that when care is delivered in a linguistically and culturally familiar setting, patients exhibit better adherence to medication [36], greater participation in self-care practices [43], and improved overall health outcomes [42,43]. Regular interactions with culturally aware healthcare providers help build trust and reduce healthcare disparities, minimizing avoidable hospitalizations and complications [31,54]. Additionally, research highlights that older migrants receiving care from community health workers who share linguistic and cultural backgrounds experience better communication and increased trust in their healthcare providers [46,47]. Integrated mental health support in home healthcare models has also been shown to reduce anxiety and depression among older migrants, contributing to a more holistic approach to health [31,45,56]

Beyond improving health outcomes, culturally competent home healthcare plays a significant role in preserving identity and fostering a sense of belonging among aging migrants [18,61,64]. Findings highlight that living spaces designed with cultural familiarity, such as incorporating traditional meals, native language entertainment, and culturally significant celebrations, enhance emotional well-being and reduce social isolation [47,54]. For migrants who may experience cultural dislocation in later life, maintaining a connection to their cultural roots through shared experiences in caregiving environments strengthens social bonds and psychological resilience [56,60]. Shared cultural experiences between caregivers and patients also help build trust, improving communication and overall care quality [54,60]. Furthermore, care models that emphasize cultural familiarity not only improve mental health outcomes but also reinforce a sense of dignity and respect for older migrants [33,45,47].

Studies further emphasize that migrants receiving care from providers who understand their cultural background report higher levels of satisfaction and emotional comfort [42,47]. Culturally tailored care models that integrate identity-preserving practices, such as celebrating cultural holidays, facilitating native language conversations, and involving family members in care, contribute to reduced stress levels and enhanced healthcare engagement [45,54]. Studies also link identity preservation to increased trust in healthcare institutions, as culturally competent care fosters deeper connections between patients and healthcare providers [31,54]. Limited funding for culturally tailored programs and a shortage of culturally trained healthcare providers contribute to ongoing disparities in health outcomes [54,62]. Additionally, insufficient investment in migrant-focused outreach initiatives further restrict access to essential home healthcare services [31,60]. Studies also highlight the importance of increased funding for culturally adapted and responsive home healthcare services and continuous cultural competence training for healthcare providers [54,56]. Additionally, flexible care models that accommodate diverse cultural needs and preferences are essential for improving service accessibility [42,47].

## 5. Discussion

This scoping review highlights the critical role of home healthcare in addressing the complex needs of aging migrants. The findings underscore significant structural and systemic challenges that hinder equitable access to care and revealed the following themes: Cultural and Linguistic Accessibility, The Role of Informal Caregiving, Structural and Systemic Challenges, and Health Outcomes and Identity Preservation.

The included studies in this review highlight significant disparities in home healthcare access and quality among aging migrants. The findings emphasize the structural barriers migrants face, including language difficulties [65], financial constraints [66], and limited awareness of available services [15]. Research also underscores the role of informal caregiving networks, with family members often bridging gaps in care due to a lack of culturally competent services [67]. Migration and aging can significantly affect psychological well-being, often leading to increased social isolation and mental health challenges [68], as well as difficulties in navigating unfamiliar healthcare systems [15]. Additionally, findings suggest that culturally tailored home healthcare models, including bilingual providers and community-based interventions, improve health outcomes and care satisfaction [65]. However, the studies collectively highlight persistent systemic challenges, such as inadequate policy frameworks and limited investment in migrant-specific healthcare programs [65], which continue to hindering equitable access to care.

The findings suggest that challenges related to language barriers, cultural misalignment, and healthcare system constraints remain key obstacles in ensuring effective home healthcare delivery for migrant populations [46,54]. Studies have also emphasized that while certain initiatives have demonstrated success in improving healthcare accessibility, their availability remains inconsistent due to structural limitations and resource constraints [31,49]. Findings also indicate that addressing these barriers requires sustained institutional commitment to culturally competent healthcare practices, targeted policy interventions, and expanded community-based support services [46,54].

The World Health Organization [69] highlights that aging migrants often face systemic injustices, including exclusion from healthcare entitlements, lack of health insurance, and restrictive eligibility criteria for government-sponsored care programs. These barriers reinforce global health inequities, with racism and xenophobia further deterring migrants from accessing proper and timely care [70]. Migrants can struggle with a healthcare system that fails to account for their cultural norms, values, or beliefs about health and illness [71]. These structural barriers are further amplified by host country policies, which may not offer adequate protection for migrant health or sufficient protection for the aging population [72]. Zghal, El-Masri et al. [73] illustrate that healthcare providers lack cultural competence, especially in the area of knowledge of non-Western cultural health practices, and this, in turn, makes migrant patients feel misunderstood. The essentiality of culturally competent healthcare providers who can demonstrate cultural sensitivity has become a pressing need [74]. To mitigate cultural and linguistic barriers, interdisciplinary cultural competency training of healthcare professionals and policies enhancing the linguistic accessibility of healthcare services, such as bilingual staff and multilingual educational materials, should be implemented [8]. Research indicates that aging migrants who receive culturally appropriate home healthcare services report enhanced psychological health, along with lower depression and anxiety levels [65,75]. Westernized healthcare, however, often conflicts with traditional healing practices, ultimately resulting in identity issues as well as cultural dissonance [76].

Informal caregiving typically comprises family members who are frequently the initial line of care for aging migrants and can provide physical assistance and emotional support, bridging the gap between healthcare providers and aging patients who face barriers to formal services [77]. Caregiving among migrants is viewed as a family duty rather than a professional responsibility in many migrant communities, which can complicate the integration of formal healthcare interventions [77]. The centrality of the informal caregiving role in maintaining dignity and autonomy among aging migrants has been highlighted and emphasized by Kim, Wister [78], specifically underscoring the significant role of family ties in social existence. Thus, training programs and initiatives to support informal caregivers are central to maximizing caregiver and aging migrants’ outcomes in home healthcare settings. The utilization of culturally suitable interventions such as family-based care models and culturally appropriate treatment approaches favorably enhances health outcomes and identity preservation [79]. Thus, healthcare systems must encompass integrative approaches that respect and maintain cultural preferences and address the medical requirements of aging migrant populations.

Many migrants prefer to age at home due to fears of cultural disconnection, inadequate religious accommodations, and limited access to culturally familiar food and services [17,18]. Despite the benefits of home healthcare in improving chronic disease management [80] and psychological well-being [68], this review identifies persistent disparities stemming from systemic failures in policy and service delivery [65]. Studies highlight that many aging migrants revert to their native language later in life, making it difficult for them to communicate their needs effectively with healthcare providers [54,60]. While informal caregivers, particularly family members, play a significant role in bridging these gaps, the burden on caregivers remains high, often leading to financial and emotional strain [52]. This review emphasizes the need for integrating informal caregiving into formal healthcare systems through targeted support, financial incentives, and culturally sensitive training programs [47,51].

This review revealed notable cultural and regional differences in attitudes toward home healthcare among migrant populations. For example, older Chinese and South Asian immigrants often prioritize family-based caregiving due to strong cultural values of filial piety, leading to lower reliance on formal services unless supported by family members or culturally concordant providers [32,38]. Somali and Middle Eastern migrants emphasized trust, cultural respect, and religious sensitivity as essential components of care, with many expressing reluctance to engage with unfamiliar or institutionalized services [46,50]. In contrast, some European migrant groups, such as Portuguese and Italian older adults in Switzerland, demonstrated greater openness to formal care services, especially when informal networks were insufficient [55]. Additionally, access and attitudes varied not only by culture but also by local healthcare systems; for instance, migrants in Stockholm benefited from ethnically profiled services more than those in other Swedish cities, influencing their comfort with formal care [50]. These differences underscore the importance of context-specific and culturally responsive models in the design and delivery of home healthcare.

Mistrust of institutional care is another prominent barrier, particularly among migrants from cultural backgrounds where elderly care is traditionally provided within the family [16,45]. This review suggests that culturally tailored home healthcare models, which integrate religious and dietary considerations, can significantly improve engagement and satisfaction among aging migrants [47,56,67]. Moreover, the findings reveal that mental health support is often overlooked in home healthcare services for aging migrants [68]. Depression and anxiety are prevalent among older adults, yet mental health screenings remain inadequate due to a lack of culturally adapted assessment tools and training among home healthcare professionals [31,43]. Expanding mental health services within home healthcare frameworks and implementing culturally informed screening tools could address this gap and improve the psychological well-being of aging migrants [54].

### 5.1. Implications

The findings of this review highlight the urgent need to implement culturally and linguistically responsive practices in home healthcare for aging migrant populations [18,61]. To improve access and service delivery, healthcare systems should integrate culturally tailored care strategies that build trust and reflect the diverse needs of patients [18,61]. Practical measures include hiring bilingual staff, using culturally appropriate educational materials, and incorporating tools such as visual aids and cue cards to enhance communication and patient engagement [46,56].

A key actionable recommendation is the provision of ongoing cultural competence training for healthcare providers, which can improve communication and reduce service inequities [31]. To ensure relevance and effectiveness, training programs should incorporate cultural humility and emphasize patient-centered values. Collaboration with informal caregiving networks such as family and community members can help strengthen support systems and complement formal care structures. However, integrating these networks into formal care models requires careful consideration of feasibility, including caregiver capacity, legal frameworks, and resource allocation [51,57]. Expanding access to ethnically profiled services and increasing public awareness of available healthcare options are also necessary to reduce disparities and build trust within migrant communities [44,50]. Training healthcare professionals in cultural humility and competence is crucial to addressing cultural insensitivity and ensuring that care aligns with patients’ needs and values [81].

Policymakers must explore strategies that recognize and support informal caregivers without overburdening them, and community organizations also play a critical role in bridging access gaps by partnering with healthcare institutions to deliver culturally informed outreach, education, and support services [82]. These programs can alleviate caregiver burden by offering training, culturally appropriate resources, and emotional support tailored to the needs of migrant families [52]. At the policy level, long-term investment is needed to embed migrant-specific needs into healthcare planning and delivery. This includes mandating cultural competence training, adopting flexible care models, and developing assessment tools that reflect cultural diversity [54,56]. Institutions must also allocate funding for interpreter services, culturally adapted materials, and transportation support to improve access [83]. Engaging community stakeholders in the co-design of home healthcare models will ensure that services are grounded in lived experience and responsive to local cultural contexts. Strengthening collaboration between community organizations, healthcare providers, and policymakers is essential to creating sustainable, equitable home healthcare systems for aging migrants [67].

Future research should focus on evaluating culturally tailored interventions and their outcomes, particularly within underrepresented groups such as Muslim and other minority migrant populations [74]. Further investigation into the specific experiences of diverse migrant populations such as Muslims and other minority groups can provide valuable insights for addressing systemic barriers and improving home healthcare delivery. Longitudinal studies on the impact of cultural competence training for healthcare providers can also inform policy and practice improvements for home healthcare for various migrant groups. Future research should assess the long-term impact of culturally tailored interventions, explore innovative care models and digital health solutions, and examine cross-country differences in attitudes toward home healthcare among aging migrants.

### 5.2. Limitations

This scoping review provides a broad synthesis of the existing literature on home healthcare experiences among aging migrants; however, certain limitations must be acknowledged. A critical appraisal to assess the quality of the included studies was not conducted, as this step is not required for scoping reviews. Consequently, the quality and rigor of the individual studies were not assessed, which may impact the strength of the synthesized findings. Additionally, while efforts were made to capture diverse perspectives, the inclusion of studies published only in English may have excluded relevant research in other languages, potentially limiting the comprehensiveness of the review.

## 6. Conclusions

Addressing cultural and linguistic barriers in home healthcare is vital to ensuring equitable access and improved outcomes for aging migrant populations. Culturally tailored care, supported by bilingual staff, culturally appropriate materials, and comprehensive training in cultural competence, can enhance trust, communication, and service utilization. Collaboration among healthcare providers, community organizations, policymakers, and stakeholders is essential to develop sustainable, inclusive models that bridge systemic gaps and support both formal and informal caregiving networks. By fostering cultural humility, improving accessibility, and prioritizing the unique needs of diverse populations, healthcare systems can create a more equitable and effective care environment for aging migrants. Continued research and policy innovation are necessary to sustain these efforts and ensure their long-term impact.

## Figures and Tables

**Figure 1 healthcare-13-00863-f001:**
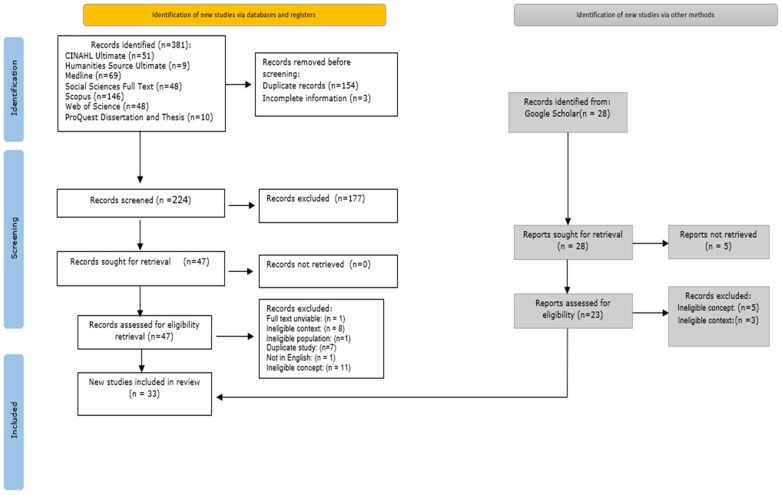
PRISMA diagram for selected studies.

**Figure 2 healthcare-13-00863-f002:**
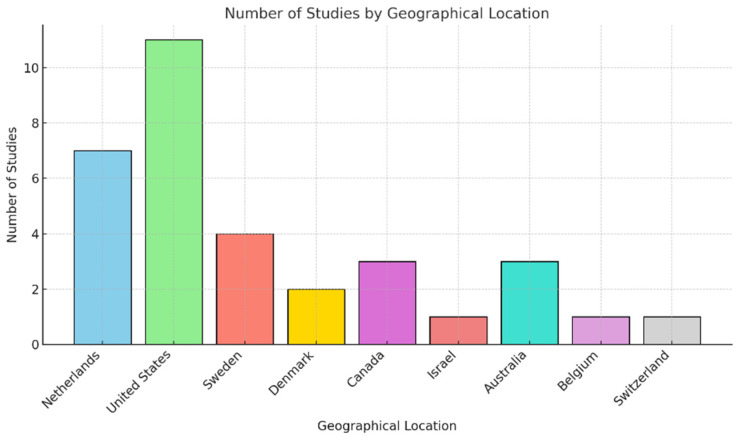
The geographical distribution of the included studies.

**Table 1 healthcare-13-00863-t001:** Criteria for inclusion and exclusion of sources.

Criteria	Included	Excluded
Population	Aging migrants and refugees (65+ years), including asylum seekers, immigrants, displaced persons, non-refugee populations.	Non-migrant older adults, younger age groups (<65), non-refugee populations.
Concept	Home healthcare services provided to older migrants and refugees, including accessibility, utilization, outcomes, and barriers.	Studies not focused on home healthcare services or not specific to older migrants and refugees.
Context	Studies conducted in any home healthcare setting, including community-based care, home nursing, and remote health management.	Studies not conducted in a home healthcare setting or addressing unrelated healthcare services.
Type of Resources	Primary research articles, review articles, theses, dissertations, reports, gray literature, published in English from 2000–2024.	Websites, blogs, commentaries, and non-peer-reviewed sources, or publications prior to 2000.

## Data Availability

The original contributions presented in this study are included in the article/Appendix A. Further inquiries can be directed to the corresponding author(s).

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
