# Peer review of "Home Healthcare Among Aging Migrants: A Joanna Briggs Institute Scoping Review"

_healthcare, 2025, doi:10.3390/healthcare13080863_

Round 1

Reviewer 1 Report

Comments and Suggestions for Authors

 The review includes my comments and suggestions.

Author Response

Reviewer comment

Response to the reviewer

Reviewer 1

Could the descriptions in the article be supplemented with

information on differences in attitudes towards home healthcare for ageing migrants in different

countries?

Thank you for this valuable suggestion. We agree that exploring cross-country differences in attitudes toward home healthcare for aging migrants would enrich the analysis. While our review focused primarily on identifying overarching themes, we have now included brief comparative insights where available in the literature to highlight how cultural, policy, and systemic contexts may shape these attitudes differently across countries. We also note this as an area for further research to better understand the influence of national contexts on service uptake and perceptions. (see highlighted section around the attitude under the discussion section.

Are there differences in attitudes to home healthcare among migrants from different cultures, nationalities, and previous places of residence (e.g., Asia, Africa, or Europe)? It would

be very interesting if the authors could try to supplement the article with the information on the

differences between home healthcare.

Thank you for your insightful comment. We agree that examining differences in attitudes toward home healthcare among migrants from various cultural backgrounds and regions of origin (e.g., Asia, Africa, Europe) is highly valuable. While our review primarily focused on common themes across studies, we have now included brief observations where the literature highlighted culturally specific attitudes and expectations. We also acknowledge this as an important area for further research to better tailor home healthcare services to the diverse needs of migrant populations. (see highlighted section around the attitude under the discussion section.

The text contains a lot of repetition of the same issues and is too long. I suggest that

the problems described should be systematized.

Thank you for your helpful feedback. In response, we have revised and streamlined sections of the introduction, discussion, and implications to reduce repetition and improve clarity. We also reorganized key issues thematically to enhance coherence and better systematize the presentation of challenges and recommendations. (see highlighted sections across the manuscript)

The structure of the text should be improved. The section of the article entitled 3.1.

Inclusion and Exclusion Criteria contains only the criteria highlighted in Table 1, as the

title of this table indicates. In contrast, the contents of sections 3.1.1 and 3.1.2 do not

deal with the issues indicated by the title Inclusion and Exclusion Criteria. These two

sections deal with other issues such as search strategy and selection and extraction of

studies. The authors have numbered them as 3.1.1 and 3.1.2, which suggests an

extension of what is covered in section 3.1.In addition, these two sections of the text

describe the researchers' workflow in too much detail, unnecessarily documenting their

every step in the research process. This could be written more concisely.

Thank you for your thoughtful comment. In response, we have removed the numbering to avoid confusion and revised the structure for greater clarity. While we recognize your concern regarding the level of detail, we believe that outlining the search strategy and study selection process in sufficient detail is important for ensuring transparency, rigor, and the credibility of the review. We have, however, streamlined the text to present this information more concisely.

Quality of English Language: The English is fine and does not require any improvement.

Thank you so much

Reviewer 2 Report

Comments and Suggestions for Authors

Reviewer 3 Report

Comments and Suggestions for Authors

Thank you for this paper and for highlighting an important issue. This review brings together a body of work that presents a valuable insight into the challenges faced by older migrants and their families, and also presents some guidance regarding how these challenges may be mitigated. I enjoyed reading this review, and wish you every success in your ongoing research.

The background is clear and presents the context for this review.

Table 1: The exclusion criteria should not be the opposite of the inclusion, so I suggest keeping non-refugee populations (as these are different to migrants) in the population row, and the exclusion criteria in type of resources. Both Concept and Context criteria are the opposite of the inclusion, and so are better left out.

The search strategy and study selection process were clear.

The results include findings related to both challenges and potential solutions to these issues.

The discussion is a synthesis of the results and reiterates the key issues and how they might be addressed.

The implications section makes a number of recommendations, particularly regarding changes that should be made by policymakers.

Minor editing errors:

Line 220: Several

Line 303: about

Lines 481-483: This sentence is incomplete. Please amend.

Lines 489-492: this sentence repeats what just went before so I suggest amending or deleting it.

Lines 505 – 508: this sentence is incomplete. Perhaps add…should be implemented, or, are necessary…

Line 514: comprising of

Line 520: The centrality of the informal

Round 2

Reviewer 2 Report

Comments and Suggestions for Authors

Thank you for addressing my concerns.